# Real-Time Monitoring Using Multiplexed Multi-Electrode Bioelectrical Impedance Spectroscopy for the Stratification of Vascularized Composite Allografts: A Perspective on Predictive Analytics

**DOI:** 10.3390/bioengineering10040434

**Published:** 2023-03-29

**Authors:** John R. Aggas, Sara Abasi, Carolyn Ton, Sara Salehi, Renee Liu, Gerald Brandacher, Warren L. Grayson, Anthony Guiseppi-Elie

**Affiliations:** 1Bioelectronics, Biosensors and Biochips (C3B^®^), Department of Biomedical Engineering, Department of Electrical and Computer Engineering, Texas A&M University, College Station, TX 77843, USA; 2Test Development, Roche Diagnostics, 9115 Hague Road, Indianapolis, IN 46256, USA; 3Media and Metabolism, Wildtype, Inc., 2325 3rd St., San Francisco, CA 94107, USA; 4Department of Biomedical Engineering, Johns Hopkins University, Baltimore, MD 21231, USA; 5Translational Tissue Engineering Center, Johns Hopkins University, Baltimore, MD 21231, USA; 6Department of Plastic & Reconstructive Surgery, Johns Hopkins University, Baltimore, MD 21218, USA; 7Department of Chemical and Biomolecular Engineering, Johns Hopkins University, Baltimore, MD 21218, USA; 8Department of Materials Science and Engineering, Johns Hopkins University, Baltimore, MD 21218, USA; 9Institute for Nanobiotechnology, Johns Hopkins University, Baltimore, MD 21218, USA; 10Department of Cardiovascular Sciences, Houston Methodist Institute for Academic Medicine and Houston Methodist Research Institute, 6670 Bertner Ave., Houston, TX 77030, USA; 11ABTECH Scientific, Inc., Biotechnology Research Park, 800 East Leigh Street, Richmond, VA 23219, USA

**Keywords:** vascularized composite tissue allografts, edema, bioimpedance, transplantation, stratification

## Abstract

Vascularized composite allotransplantation addresses injuries to complex anatomical structures such as the face, hand, and abdominal wall. Prolonged static cold storage of vascularized composite allografts (VCA) incurs damage and imposes transportation limits to their viability and availability. Tissue ischemia, the major clinical indication, is strongly correlated with negative transplantation outcomes. Machine perfusion and normothermia can extend preservation times. This perspective introduces multiplexed multi-electrode bioimpedance spectroscopy (MMBIS), an established bioanalytical method to quantify the interaction of the electrical current with tissue components, capable of measuring tissue edema, as a quantitative, noninvasive, real-time, continuous monitoring technique to provide crucially needed assessment of graft preservation efficacy and viability. MMBIS must be developed, and appropriate models explored to address the highly complex multi-tissue structures and time-temperature changes of VCA. Combined with artificial intelligence (AI), MMBIS can serve to stratify allografts for improvement in transplantation outcomes.

## 1. Introduction

Clinical outcomes following vascularized composite allotransplantation have demonstrated that vascularized composite allografts (VCA) are a viable treatment option for patients suffering large tissue defects or loss of limbs and for whom there are no conventional reconstructive options [1]. Unlike solid organs, these complex anatomical structures contain multiple tissue types, including skin, bone, fat, muscle, connective tissue, and nerves, and must be appropriately preserved and/or immunologically conditioned on their path from donor to the recipient [2]. To date, more than 120 such transplants have been performed worldwide, including upper extremity and face transplants [3,4]. Vascularized composite allotransplantation improves the quality of life of patients and may accrue considerable economic benefits for both patients and communities [3,5].

The logistical challenges of VCA transplantation and the subsequent need for lifelong immunosuppression restrict VCA accessibility and motivate the need to improve transplantation outcomes. Central among these is the development of VCA preservation technologies and VCA conditioning protocols. Presently, standard organ preservation employs cold storage conditions, under which irreparable VCA ischemic damage develops within 4–6 h [6] and reduces the rate of successful transplantation [7]. Consequently, geographic distance is directly correlated with ischemia time and negatively correlated with transplantation outcome [8]. To extend viable transfer time and expand VCA availability, advanced methods of VCA preservation, continuous physiological status monitoring, VCA stratification, and prognostic assessments based on relevant performance data are needed. Static cold storage has been shown to present major limitations for VCA [8]. In response to these limitations, our team has reported on the design, development, and application of multi-parametric bioreactors for the functional preservation of vascularized composite allografts [9]. Such systems have been imbued with the capability to support ex vivo electrical stimulation to promote muscle regeneration and reduce muscle atrophy [10,11,12] while monitoring physiological biomarkers [13,14].

Over the last few decades, ex vivo machine perfusion systems have emerged as a viable option for prolonging organ preservation time beyond the limits of cryopreservation [15]. Such systems allow simultaneous perfusion and preconditioning with immunomodulatory molecules [16]. This lowers the probability of transplant rejection while reducing treatment dosage for patient immunosuppression [8]. Furthermore, preservation with immunomodulation may enable the use of VCAs that would otherwise be deemed unsuitable for transplantation due to immunological factors, including alloreactive immune responses [6,17,18]. Presently, invasive assessment of VCA viability includes tissue biopsy and histology, with inflammatory infiltrates used as markers of graft injury or rejection onset. Alternatively, intra-graft inflammatory cytokine profiling and gene expression profiling have been employed to establish quantitative metrics for graft viability and predict the onset of acute rejection. Edema, fluid accumulation within the tissue bed, is a key clinical indicator of VCA viability. Non-invasive monitoring of perfusate parameters include measurement of pH, flow rate, pressure, temperature, and gas composition combined with biochemical assays of the perfusate to measure damage-associated markers such as creatine kinase and myoglobin [19,20]. Biochemical markers such as glucose, potassium, lactate, calcium, and oxygen have also been used to assess VCA transplant viability during preservation [21] and, along with acidosis, may be the basis for the development of indwelling bioanalytical biochips for the monitoring of allografts [22,23]. Concurrently, indicators of ischemia-reperfusion injury in VCA have been identified to monitor damage to tissues during reperfusion [6,21]. Biochemical sensors, such as those proposed for the management of the hemorrhaging trauma patient, may provide real-time, temporal feedback for key metrics in transplant viability, including metabolism and the accumulation of toxic metabolites [22]. This paper presents a perspective on the use of Multiplexed Multi-electrode Bioelectrical Impedance Spectroscopy (MMBIS) [24] as a suitable technique for assessing clinically relevant edema. Furthermore, this method has the potential to provide the spatial and temporal resolution required for continual VCA monitoring during preservation. When combined with AI, MMBIS may enable prognostic stratification of VCA to increase successful outcomes following transplantation. VCA, unlike solid organs, is particularly challenging because of the considerable tissue heterogeneity of the allograft. Moreover, when compared to solid organ transplantation, acute rejection rates are approximately six times greater. This necessitates vastly different pre-transplantation protocols with aggressive immunosuppression and, quite often, incurs loss of graft viability. The integration of MMBIS biophysical data, bioanalytical physiological data, and clinical expert data, once successfully applied to preserved allografts, may potentially be applied to post-transplant rejection and tolerance evaluation and scoring.

Edema, the retention of excess fluid within tissues, is an important clinical indicator of tissue injury and is commonly used as an indicator of graft injury or rejection [25]. As a defined physical phenomenon, edema has the potential for continuous, real-time monitoring in the assessment of allograft preservation and transplantation viability. Historically applied to VCA histopathology, the multi-modal Banff criteria is a semi-quantitative method that considers the extent and localization of inflammatory infiltrate, apoptosis, and necrosis [25]. The production of immuno-stimulatory molecules and infiltration of macrophages into muscle and skin [26] accompany ischemia-related graft injury that triggers inflammatory signaling and increases cellular infiltration, resulting in edema. Edema is correlated with poor VCA outcomes [4,27]. During preservation, perfusion with immunosuppressive drugs may serve to mitigate rejection but imposes highly stringent protocols with complex drug regimens [4]. Measuring edema is one approach to monitoring the progression of acute rejection that enables timely and appropriate intervention. Bioimpedance analysis permits quantitative assessment of tissue based on intrinsic electrical properties and is highly discerning with regard to the pathophysiology at the molecular, cellular, tissue, and organ level according to the applied frequency range [24].

## 2. Bioimpedance for Non-Invasive Tissue Evaluation

Bioelectrical impedance (BI), bioelectrical impedance spectroscopy (BIS), and bioimpedance tomography (BI/BIS/BIT) are promising tools for evaluating VCA in the context of preservation and monitoring of edema. Impedance is the frequency-dependent electrical characteristic of a material and is a measure of its opposition to an alternating electrical current. Impedance is typically measured in a four-electrode set-up. A contacting outer pair of electrodes serves as the current source, while sink electrodes and an inner pair serve as sense electrodes for measuring the ensuing voltage drop, as shown in Figure 1A. The impedance of the tissue under test (TUT) is sometimes presented as Bode (magnitude and phase vs. frequency) or Nyquist (imaginary vs. real component) plots shown in Figure 1B. Impedance data are often interpreted by defining an equivalent circuit that best fits and realistically represents the acquired frequency-dependent data, with each circuit component reflecting a real, tangible physicochemical feature or process occurring within the TUT (Figure 1C) [28,29]. The application of frequency-dependent impedance to the simplified Cole Model for a homogeneous isotropic tissue material under test yields the following equations for the impedance (*Z*), the resistance (*R*), and reactance (*X*),
(1)$$Z\left(\omega \right)=R_{\infty }+\frac{R_{0}-R_{\infty }}{1+{\left(j\omega \tau \right)}^{\alpha }}$$
(2)$$R\left(\omega \right)=R_{\infty }+\frac{\left(R_{0}-R_{\infty }\right)\left(1+{\left(\omega \tau \right)}^{\alpha }\cos\left(\frac{\alpha \pi }{2}\right)\right)}{1+2{\left(\omega \tau \right)}^{\alpha }\cos\left(\frac{\alpha \pi }{2}\right)+{\left(\omega \tau \right)}^{2\alpha }}$$
(3)$$X\left(\omega \right)=-j\frac{\left(R_{0}-R_{\infty }\right){\left(\omega \tau \right)}^{\alpha }\sin\left(\frac{\alpha \pi }{2}\right)}{1+2{\left(\omega \tau \right)}^{\alpha }\cos\left(\frac{\alpha \pi }{2}\right)+{\left(\omega \tau \right)}^{2\alpha }}$$

The Cole Model consists of low-frequency resistance (*R*_0_), high-frequency resistance (*R*_∞_), a shape parameter (α) associated with dielectric dispersion, and a time constant (*τ*) [30,31].

**Figure 1 bioengineering-10-00434-f001:**
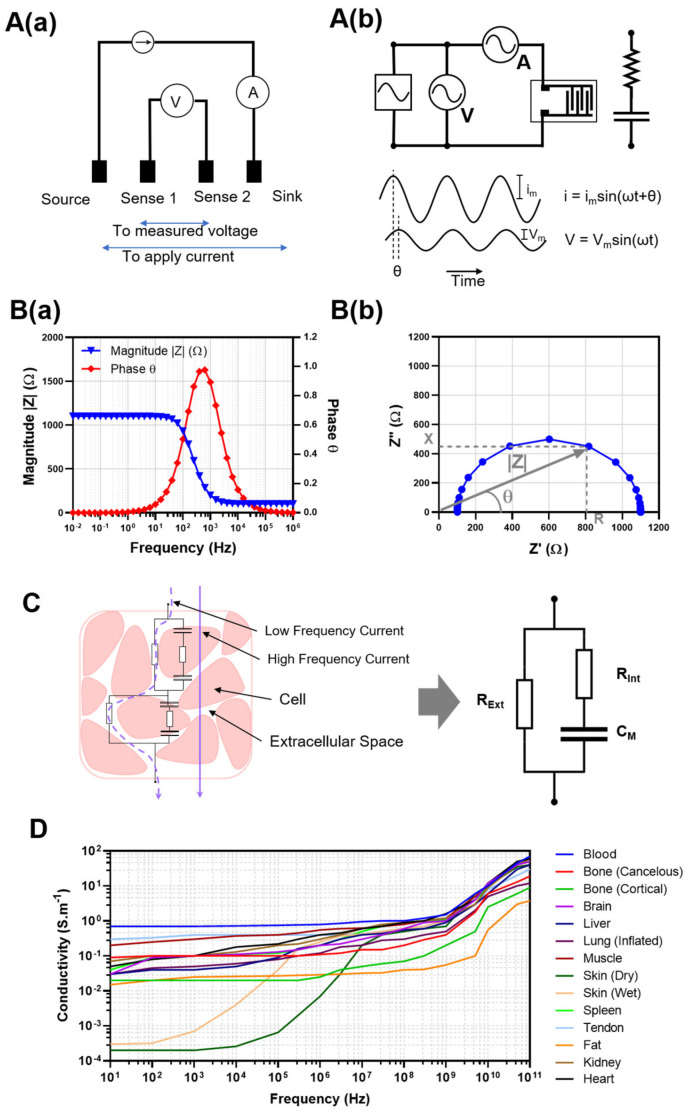
**A**(**a**) A four-point impedance/resistance measurement set-up. The two outer electrodes are used to apply a current, and voltage is measured from the two inner electrodes. **A**(**b**) Generalized description of multispectral impedance spectroscopy applied to an equivalent circuit. The alternating current with a magnitude of i_m_ and frequency of *f* (ω = 2πf) applied to the element results in a voltage with a magnitude of V_m_ and a phase of θ. **B**(**a**) Bode plot: magnitude(|Z|) and phase (θ) versus frequency indicating the frequency-dependent impedance. **B**(**b**) Nyquist plot: real (Z′) and imaginary (Z″) parts of impedance. The size of the border from the center shows the magnitude. |Z| of impedance at each point and the angle with the *x*-axis shows the phase, θ. The perpendicular projections of the Z border onto X- and Y-axes show resistance (R) and Reactance (X), respectively. (**C**) A schematic illustration of the frequency-dependent path of the dominant current through the tissue bed, showing low and high-frequency current pathways. (**D**) The conductivity of different tissue over a frequency range of 10 Hz–100 GHz. The graph is reconstructed using data from Gabriel et al. [32].

Quantitative assessments by bioimpedance analysis access the multiple length scales of molecular, sub-cellular, cellular, tissue, and organ according to the specific frequency or frequency range selected. The measurement is based on the intrinsic electrical polarization responses of water molecules, hydrated ions within the interstitial spaces, the potentials across cell membranes, the presence of ion channels within those membranes, the permittivity of the cell types under test, and the 4-D arrangement of cell types within a complex, heterogeneous tissue [24]. The frequency range is characterized by the alpha (α), beta (β), delta (δ), and gamma (γ) dispersions that correspond principally, but not exclusively, to the migration and steaming of ions within the interstitial space (α ≈ Hz–kHz), the rotational dynamics of polar proteins and membrane potentials (β ≈ kHz–MHz), the hydrogen-bonded water bound to biomolecules (δ ≈ MHz–GHz), and the polarization dynamics of water (γ ≈ 10 GHz) [24].

BI may be implemented as temporal measurements at a single frequency (impedimetry) or multiple frequencies (multi-frequency spectral or BIS), intended to reveal relative changes in TUT over time. This approach assumes that the physicochemical reality of the equivalent circuit model is time-invariant or that the time scale of these measurements is short relative to possible physiological or pathophysiological changes. Sub-cellular fragments, cells, tissues, and organs are real systems rather than perfect resistors, capacitors, and inductors and are, therefore, leaky equivalents of their solid-state counterparts. Current through a TUT is established in a frequency-dependent manner (Figure 1C). At low frequencies, the current is mainly established in the extracellular space. As the frequency increases, coupling into the leaky capacitor of the cell wall allows current to pass through the dielectric membrane of the cells and into the intracellular space. The equivalent circuit that best describes this behavior (Figure 1C) is comprised of a resistance (R_Ext_) representing the extracellular component in parallel with a serial capacitance (C_M_) and resistance (R_Int_) representing the cell membrane and intracellular component. This simple yet descriptive model for the electrical behavior of tissue in response to an alternating current is widely used in BI-based monitoring devices and underscores the importance of frequency range when using BI/BIS. Figure 1D shows the conductivity as a function of the frequency of a number of tissue types over a frequency range of 10 Hz–100 GHz reconstructed from published data [32]. Judicious choice of frequency or frequency range thus determines the meaningfulness of the BI/BIS measurement. In studying the preserved and transplanted solid organ rat kidney model, Genescà et al. [33] used the Cole model parameters to clearly differentiate between two groups, distinguished by their BIS behavior during reperfusion. They identified the *α* parameter, the width of the bioimpedance dispersion, as a feature that is both objective and easy to allow real-time, continuous monitoring to detect and monitor ischemic damage during rat kidney preservation and “could be a promising method to assess graft viability in the future.” Additionally, τ, the characteristic time constant of the dispersion, was used to differentiate between the two groups during the short preservation times, ~24 h. Both *α* and τ implicate the increased volume of ischemic cells for its impact on reduced interstitial volume and increased cell membrane surface area.

Commercially available, USFDA-cleared bioimpedance devices and systems have been developed and deployed for clinical use [24]. Amongst them is Baxter-toSense’s CoVa Monitoring System 2, a remote patient monitoring necklace that tracks vitals and cardiac fluids relevant to congestive heart failure (CHF), chronic obstructive pulmonary disease (COPD), hypertension, and renal failure. Cardiac fluids are tracked by bioimpedance plethysmography (10 kHz to 500 kHz) and serve as an early indicator of heart failure. The Philips Biosensor BX100 is a wearable system deployed as a single-use, 5-day wearable patch that uses thoracic bioimpedance (20 kHz to 100 kHz) to monitor the respiration of patients. The uCor3 by ZOLL Medical Corp. is similarly deployed as a patch for thoracic RF bioimpedance (0.5–2.5 GHz) for heart failure monitoring.

### 2.1. Application of BI to the Measurement of Clinical Edema

BI measurement at a single frequency (e.g., 5 kHz, 50 kHz, or 86 kHz) from one pair of contacting electrodes is the simplest form of bioimpedance which does not require sophisticated instrumentation and subsequent complicated data analysis. The data are used to produce a map of the permittivity (ε), resistivity (σ), or conductivity (ρ) of TUTs. Increasing the number of electrodes used beyond a single pair provides the ability to use individual pairs for current injection and voltage sensing. Increasing the number of electrodes tends to enhance the spatial resolution of the map, although too many electrodes could result in a degraded signal [34]. This technique has great potential in monitoring biological events where the permittivity of the TUT is affected [35]. Consequently, BI has been widely developed and applied to provide superior outcomes in clinical situations that involve the accumulation, loss, or movement of liquid or air, such as in assessing lung capacity and function, the cardiovascular system, edema, and lymphedema [36].

Zink et al. [37] compared body-segmented BIS data with traditional clinical diagnostics in a study of fluid accumulation and status during cardiac recompensation. Because clinical signs of volume overload, such as breathlessness and peripheral edema, vary widely, the authors compared “whole-body” and “foot-to-foot” bioimpedance values, which were lowered for cases of peripheral edema, and “hand-to-hand” and “transthoracic” bioimpedance, which were lowered for cases of central edema. Edwick et al. [38] have demonstrated the use of BIS as a valid, reliable, and sensitive technique in the measurement of extracellular fluid volumes, and hence edema, following acute hand burn injury. Such as our purpose in VCA monitoring, they emphasized the importance of changes in edema to augment clinical decision-making. The authors extracted the impedance of the intracellular component from a fitted Cole model to calculate the extracellular fluid (ECF) volume. These values were then compared with values obtained from the water displacement method, where the swollen hand was placed in a liquid bath, and the volume displaced was measured, resulting in a Pearson’s correlation of r = 0.793 (*p* < 0.001). Crescenzi et al. [39] used BIS to assess the temporal accumulation of tissue water and were able to distinguish lipedema from Dercum’s Disease with an emphasis on identifying the early stages of lipedema. The authors found a reduced R_0_ of the Cole model to be associated with the higher volume of swollen legs of patients with more severe lipedema. This corresponded with the clinical manifestations of pitting upon palpation, and it was assumed that this was the result of a greater accumulation of fluid within the interstitial spaces of the tissues. The reduced impedance and increased interstitial fluid volume were suggested to occur because of an increase in sodium and associated counter-anion content in the interstitial space [40].

Assessment of edema using multi-frequency bioimpedance measurement provides direct insight into fluid accumulation as the tissue is perfused [41]. Low-frequency current has high penetration depths and travels deep into the tissue while following the path of the extracellular fluid. At high frequencies, the injected current overcomes the cell membrane capacitance and detects intercellular fluid. The combination of intracellular and extracellular fluid measured by the differing frequencies applied can be analyzed to estimate the extent of edema, potentially requiring complex instrumentation and data interpretation. This resolution typically comes at the cost of more complex and sophisticated instrumentation and data analysis. Nonetheless, York and colleagues have shown single-frequency impedance and impedance spectroscopy yielded similar estimations for edema in the legs and arms of patients afflicted with lymphedema [42]. Measuring BI at two frequencies instead of one serves to eliminate irrelevant data through some form of normalization, e.g., subtraction or rationing [43,44], and enhances the visualization of the object of interest (e.g., extracellular fluid). The ratio of impedance collected at two frequencies, 100 kHz (high) and 5 kHz (low), allowed assessing edema in the ankle [44]. A normal ankle joint could be distinguished from an injured one using the corresponding impedance ratio for each condition, even when the subject’s ankle moved in different positions. Furthermore, the impedance ratio remained constant when testing each leg of a healthy subject, while it varied in a subject with edema. Lyons and colleagues performed a BIS assessment of the edema index in 359 outpatients with heart failure using the commercially available InBody 520 scale at 5, 50, and 500 kHz [45]. By using multiple frequencies in multiple locations of the body, intracellular and extracellular fluids were measured by the device’s algorithm and an edema index was calculated using a ratio of extracellular fluid impedance to total body water. Patients were stratified into those with low (≤0.39) and high (>0.39) edema indices to provide insight for outcomes including death, urgent transplant, or ventricular assist device implantation. After a 2-year follow up, the authors showed that patients with a higher edema index had more edema-related complications, indicating the feasibility of using multi-frequency BI as a prognostic metric.

There are other methods to measure edema. Among them are the resect and weigh to dry weight method or the fluid volume displacement method. The former is a highly invasive method that destroys both the sample and further traumatizes the VCA. The latter requires the allograft to be “dunked” into a fluid bath and the volume displacement of fluid measured. Both methods are highly disruptive to a perfusion system. In the work of Strand-Amundsen et al. [46], they used bioimpedance to study the temporal changes in the electrical properties of the multilayered small intestine, both in vivo and ex vivo, during ischemia development. The authors found significant differences (*p* < 0.0001) between the evolving electrical parameters as a function of ischemic time development when comparing measurements performed in vivo vs. ex vivo. However, there were notable similarities in the trends, both trending downward, likely due to edema, with the ex vivo data showing a low-frequency rise for about 3 h prior to beginning its downward trend. The temporal evolution of the phase angle, ϕ, was similarly significantly different (*p* < 0.0001) when comparing the baseline condition and the development of ischemia in both the in vivo and ex vivo models. It is noteworthy that these differences are more strongly frequency dependent and typically occur in the 1–7 h time frame. 

Peterson et al. [47] used bioimpedance to study total organ edema of porcine lungs undergoing normothermic extravascular lung perfusion (EVLP). They used Steen, an asanguinous and hyperosmolar solution, normal saline (0.9%), and cell culture media to determine if bioimpedance was associated with vascular lung water and lung injury severity. Overall, increases in pulmonary edema and extravascular lung water have a direct negative correlative effect on lung organ quality. Steen is known for its protective effect on the pulmonary microvascular endothelium [48]. Saline and Cell culture media both induced pulmonary edema and lung injury. Steen, on the other hand, reduced lung injury and extravascular free water over the duration of normothermic EVLP. They found that bioimpedance could provide a quantitative organ assessment that would allow for more accurate pre-transplant clinical decisions. In the recent work of Hou et al. [30], they studied the temporal bioimpedance of resected segments of the highly layered human small intestine. They found that the normalized resistance ((R_0_ − R_i_)/R_0_) × 100 = P_y_) initially decreased and subsequently increased before decreasing over a 10 h period of monitoring. The time interval where the P_y_ value returned to its maximum value was found to be consistent with reported viable/non-viable limits based on histological analysis. In this case, ischemia was associated with the internal swelling of cells and changes in the ratio between intracellular and extracellular fluids, edema. In the related work of Hou et al. [49], dielectric relaxation spectroscopy was used to assess the intestinal viability of healthy, ischemic, and re-perfused segments of the intestine in a porcine mesenteric ischemia model. The bioelectric parameters of dielectric constant and conductivity showed clear differences between healthy, ischemic and re-perfused intestinal segments. This confirmed the value of dielectric parameters in characterizing different intestinal conditions likely based on edema. They then employed machine learning models to classify viable and non-viable segments based on frequency-dependent dielectric properties of the intestinal segment. The combined approach of dielectric relaxation and trained machine learning is speculated to allow the surgeon to make an accurate evaluation by performing a single measurement on an intestinal segment where the viability state is questionable. This then augments the surgeon’s difficult decision regarding resection segments and resection margins. Moreover, the early stages of the temporal changes in bioimpedance measured during ischemia produce a pattern of edema that is different in the porcine and human small intestine.

### 2.2. Application of Electrical Impedance Tomography (EIT) to Edema Measurement

Single-frequency BI monitoring has been the basis of several classical tomography techniques since the 1950s and has been developed to produce a 3D image or heat map in a technique referred to as electrical impedance tomography (EIT) [50]. In this technique, an array of electrodes is spatially arranged on the TUT, and impedance is sequentially measured from each pair. Image reconstruction algorithms are applied to impedance data to construct a 2D/3D image of the TUT, e.g., the chest, in the form of a heatmap, distinguishing features of differing electrical characteristics. While single-frequency EIT is the dominant mode of impedance measurement in EIT, increasing research has been dedicated to improving the resolution and specificity of the technique using multiple frequencies, i.e., spectral techniques. EIT has been used to monitor the heart, lung, breast, brain, and other parts of the body and may substitute—or complement—other imaging modalities such as CT and µCT [51,52]. EIT allowed the diagnosis of fluid accumulation within the long and distinguished edema from atelectasis in animals with acute respiratory distress syndrome [53]. Presently, EIT has poor reproducibility and limited diagnostic applicability. However, advantages, including rapid data acquisition and hardware portability, justify further development of this monitoring technique in order to bring to the clinic a non-invasive, rapid, and low-cost imaging modality that enables instantaneous and unobtrusive tissue assessment [54]. Much of the effort in the field is currently directed toward improving and expanding image reconstruction algorithms while removing background signals in software and miniaturizing current/voltage sources that support a wider frequency range in hardware [55].

### 2.3. Potential for Applying BI/BIS to Ex Vivo VCA Preservation

BI/BIS enables the detection and monitoring of changes in tissue permittivity due to the accumulation of fluid or the movement of blood/fluid within tissues. These techniques have the potential for ex vivo monitoring of edema during VCA perfusion within a preservation bioreactor as well as in vivo to detect incipient allograft rejection following transplant reperfusion. As extensively discussed, frequency selection is the most important aspect of this technique’s practicality [24]. In single-frequency mode (BI), the simplicity of its instrumentation requirements offsets the effort required to find the optimal frequency and capture sufficient information during BI. If monitored at the proper frequency, the absolute BI data reflects the parameter of interest, which, in the focus of this review, is the spatiotemporal development of edema. This approach provides the advantage of nearly instantaneous updates of an empirically derived heat map with a dynamic variation of the image, similar to a movie. The use of BI has a major disadvantage in that edema develops over time during preservation and so identifying an optimal frequency that is applicable over the time course of monitoring is challenging. BIS, on the other hand, necessitates the application of a range of frequencies within which the temporally evolving optimal frequency should reside. However, wide ranges of frequencies, particularly toward low values, requires long data acquisition times between pair-wise electrode measurements, leading to an abundance of data but for a system that may experience temporal drift between the first and last multiplexed measurement.

A recent study using multi-electrode, multi-frequency bioimpedance for monitoring temporal biological events resulted in the development of an electrical cell stimulation and recording apparatus (ECSARA) [56]. ECSARA used a multiplexing algorithm to apply electrical stimulation and concurrently records bioimpedance data from 24 electrode pairs within a 24-well electro-culture ware. The system has the potential to be adapted for simultaneous tissue stimulation (e.g., to promote innervation, reendothelialization, or for in vitro stimulation of skeletal muscle) and bioimpedance measurement over an area equal to the multiplexed electrode footprint, yielding a conductivity map representing tissue electrical properties. A modified schematic of the system configuration that can be applied to VCA perfusion monitoring is shown in Figure 2. However, the challenge with multiplexed, multi-electrode, multi-frequency approaches is that images obtained do not reflect a “single point in time.” Occasionally, considerable time (~4 min for a 10 mHz–1 MHz sweep) is required to obtain the raw data, process useful data elements, and export these to an image presentation algorithm. The key is for image acquisition update time to be small compared to the time scale of the dynamic physiological events being monitored. Such is the case with tissue edema of a perfused rat abdominal wall VCA [9]. Clinically, initial transplant-associated edema is resolved over a 7–10-day period, after which rejection-associated edema becomes evident. In previous work, large animal ex vivo VCA has been perfused for several hours, with a maximum of 24 h [6,20]. Considering the protracted time scale of VCA perfusion and edema development, multi-frequency BIS remains a valid monitoring option, although low frequencies (<10 mHz) should be excluded to maintain high temporal resolution. Since the fundamental basis of BI/BIS in tissue detection is electrical permittivity, the technique is, in principle, capable of discriminating edema-induced changes within layered or non-layered, heterogeneous tissue [57]. However, tissues that comprise VCA have close electrical properties and are distributed non-homogenously within the allograft. Although BI/BIS can feasibly monitor dynamic fluid accumulation and potentially its flow as a cumulative parameter, pinpointing such changes with regional specificity requires Focused BIS (FBIS) [58]. FBIS requires triangulation of electrodes with overlapping voxels of interrogation to increase sensitivity to changes within the targeted zone, hence a zonal focus. Focused BI/BIS allows the identification and monitoring of specific zones within a complex organ or tissue (e.g., cirrhosis of the liver), the progress of a solid tumor, or regional distribution of edema within a VCA. The research to improve and optimize the measurement towards higher spatial resolution is ongoing [59]. While there may be interest in the underlying causes of edema and the molecular etiology of changes in the bioimpedance associated with the development of edema, the present perspective seeks only to holistically stratify the VCA based on the fusion of pathophysiology analytical data, clinical expert data, and MMBIS data to yield stratified performance. The authors have employed similar approaches in the development of a Hemorrhagic Trauma Severity scoring system [23] for the hemorrhaging trauma patient [60]. 

### 2.4. Potential for ML/AI in Allograft Stratification

Following resection from the donor, allografts generally undergo preservation that may include perfusion, usually within a bioreactor [9,15,61], physiological status monitoring bioassays [62], and immunomodulatory therapies [16] to minimize rejection due to ischemia-reperfusion injury following transplantation [4]. Even so, short preservation times (4–6 h) and the absence of rationally defined allograft viability stratification produces the well-documented bias toward extremity amputations: 10,000 (military and civilian) in 2010 compared to 100 patients worldwide have benefited from VCA, hand or face transplants [63]. Macroscopic rejection and histopathological scoring of skin-associated allografts based on the Banff criteria [26] has established Grade 0—No signs of rejection, Grade 1—Erythema; Grade 2—Erythema and Edema, Grade 3—Epidermolysis, Grade 4—Mummification and Necrosis with elevated cytokines IL-4, TNF-α, and IL-12p70 being able predictors of skin rejection before evidence of histopathological alterations [64].

Allograft viability stratification prior to transplantation may be enabled by a quantitative fusion of the multi-modal bioanalytical/bioassay and biophysical data with clinical expert data via machine learning/artificial intelligence algorithms [62]. Stratification of allografts during preservation for success prior to transplantation is possible via a scoring metric such as that employed in post-transplant rejection scoring. Here we propose four levels of stratification for allografts during preservation. Thus, 0 = no ischemic damage (no edema), 1 = mild ischemic damage (mild edema, no erythema), 2 = moderate ischemic damage (edema and erythema), and 3 = severe ischemic damage (epidermolysis and/or necrosis). Such approaches may be evaluated by passing the discrete data to the classifier or by passing derivative data, such as biomarker scores, to the classifier [60,65]. Among the relevant scores is the Hemorrhage Intensive Severity and Survivability (HISS) score, which aggregates molecular metabolite data into a delineation of hemorrhagic shock states [23], and the emerging inflammatory status score, which aggregates molecular indicators of the inflammation status into a delineation of inflammation states [66]. By applying machine learning/artificial intelligence algorithms to combine expert knowledge and bioanalytical/bioassay data that were continually gathered from the VCA (multi-modal data) before, during, and after perfusion and/or transportation of the allograft, an efficacious scoring metric may be developed [65]. 

Specifically, temporal changes in pressure, temperature, glucose, lactate (lactatemia), pH (acidosis), [PO_2_], K^+^ (hyperkalemia), cytokine profiles, and multiplexed, multi-electrode bioimpedance spectroscopy can provide holistic and continuous data to monitor the quality of allografts. Elevated levels of pro-inflammation and anti-inflammatory cytokines are expected to accompany the trauma of resection of a VCA [67,68]. Unfortunately, there is presently no consensus on what constitutes a suitable and specific biomarker profile for successful allograft preservation. Moreover, such profiles are not linked to the potential for successful transplantation outcomes. Key molecular parameters such as C-reactive protein (CRP), the cytokines: IL-1α, IL-1β, IL-4, IL-6, IL-8, TNF-α, IL-12p70, IL-10, and IL-18 are each implicated in the inflammatory status following trauma [69] with IL-18, IL-1β, and IL-4 being key indicators of transplant rejection [64]. The cytokines IFN-γ, IL-1β, IL-2, IL-1α, IL-1Rα, and IL-6 have been identified as biomarkers for VCA [70]. Among the other small molecules of great relevance to allograft stratification are iNOS, ROS, Caspase-3, HMGB-1, and Necrostatin-1. The challenge is identifying readily measurable, decision-worthy small molecules that support predictive analytics [70,71]. These bioassay, biomedical sensor, and possibly biosensor [72] data may serve as fused inputs into stratification algorithms [73] such as multi-class linear support vector machines (SVM-L), ensemble bagged decision trees (EBDT), artificial neural network (ANN) classifiers, or possibility rule-based function approximations (PRBFA) classifiers, resulting in a single actionable output score [23] (Figure 3).

ANN with Bayesian Regularization (ANN:BR) tunes incoming animal data sets [74]. The inputs are turned into a linear model (*ωx* + *b*), where ωx is the matrix multiplication of weights (*ω*) and inputs (*x*), and *b* is the bias. The scores obtained from this step are fed into the Softmax activation function (Equation (4)) to introduce non-linearity and converts them into probabilities.
(4)$$\sigma {\left(z\right)}_{j}=\frac{e^{z_{j}}}{\sum_{k=1}^{K}e^{z_{k}}}\;for\;j=1,\;2\ldots ,\;k$$

Softmax function maps the set of outputs onto inputs. In this case, there are five outputs (0 through 4) which, when passed through the Softmax function, become distributed according to probability (0, 1), defining the most probable occurrence or classification for a particular output.

PRBF with Function-Approximation (PRBFA) [75] handles uncertainty in expert knowledge. The degree of belonging of an instance to the $k^{th}$ class may be characterized by $u^{k}\in \left[0,\;1\right].$ Under the possibility theory [75,76], $u^{k}$ is the measure of possibility that the given data point belongs to the class of score *k* and the following representation holds for the set of possibilistic classes assigned to the $i^{th}\;$ instance: (5)$$u_{i}=\left(u_{i}^{1},\;u_{i}^{2},\ldots ,u_{i}^{c}\right)\;\forall u_{i}^{k}\in \left[0,\;1\right]$$

In Equation (5), *c* is the number of scores defined for the problem, i.e., in this case, five VCA allograft grades of 0 (no adverse reaction); 1 (edema); 2 (erythema); 3 (epidermolysis or desquamation); and 4 (necrosis) based on expert opinion borne from experience and knowledge. Unlike the probabilistic labels, the values of the vector, $u_{i\;},$ do not have to sum to unity. Instead, each parameter takes a value ranging from 0 to 1. The classification scheme proposed by Nazmi and Homaifar [75], namely, a possibility rule-based classifier using function approximation (PRBF), employs this definition of a class assignment and trains a rule-based evolutionary model that, given a data point, predicts the degree of possibility to which the multi-modal VCA data set belongs to each of the possible classes.

Performance, Cross-validation, Data Set Size. In general, the performance of a multi-class classification can be measured using accuracy, precision, and an F–score [77]. A confusion matrix plot, comprising values corresponding to true labels and predicted labels, is often used to evaluate the quality of each classifier [78]. The values in the major diagonal of the confusion matrix serve to determine how well the classifier has performed. Accuracy, when used as a performance metric to report the prediction performances of each classifier, is obtained from the major diagonal elements of the confusion matrix as follows (Equation (6)),
(6)$$Accuracy=\frac{\#\;Correct\;predictions}{\#\;predictions}=\frac{\sum \;of\;elements\;in\;the\;major\;diagonal}{\#\;of\;elements}$$

Sparse data presents a challenge and may be addressed with data being arbitrarily split into equal sets for the training of multiple models. A 5-fold cross-validation [79], when employed [80,81], helps with using all available data for model training and so establishes robust predictions. The adequacy of the multi-modal data size is established as the minimum needed to stabilize the validation accuracy. The adequacy of the number of clinical expert classifications and the prediction for the size of the bioanalytical/bioassay data set needed to achieve a test accuracy of 0.95, 0.99, and 0.999 with the predicted number of experts necessary to achieve that accuracy is generally arrived at using a regression model fit and application of predictive modeling.

### 2.5. Challenges of BI/BIS Interrogation

The use of BI/BIS-based techniques provides an array of challenges that must be addressed in order to promote proper measurement and data collection: (1) the frequency or frequency range to be used, (2) the choice of electrode material, (3) the size of electrode features (resolution/voxel size), (4) maintaining proper electrical contact with the TUT, (5) the non-specific nature of BI/BIS measurements, (6) excitation signal injection method and (7) instrumentation footprint. Defining the frequency to be used in measurement must first be governed by the parameters to be measured from the TUT (i.e., α, β, δ, γ) [24]. 

Given that the VCA is subject to major changes in its impedance spectral characteristics during perfusion and possibly swelling and that temporal drift in the contact impedances between electrodes and tissue do arise, a major limitation of the single frequency approach is found in the question, “what is the proper frequency?” [82]. MMBIS allows the measurement and tracking of the characteristic time constant of the TUT. This may be used to dynamically update the frequency range, thereby balancing data points on either side of the changing time constant, a feature not presently employed by any system we have reviewed. However, contact impedances from physical electrodes often confound the impedance of the TUT; therefore, a judicious choice of electrode materials and conductive adhesive is necessary [56]. In addition, low interrogation frequencies result in long-time scales for individual data points (reduced temporal resolution). This limits the ability to measure physiological events that alter ion concentration or ion mobility, trans-membrane permeability, and molecular polarizability within the tissue relative to the time scale of an impedimetric or spectral measurement. In addition, real tissue is not homogeneous and isotropic and is therefore subject to spatiotemporal variations. This greatly impacts the volume element or voxel that is sampled by the contacting electrodes and defines the sampling resolution. In order to reduce contact impedances and enhance signals from the skin, electrode surfaces are often covered with a conductive gel. Long-term contact with the VCA under perfusion conditions may alter the gel’s electrical properties and result in baseline drift to potentially confound results [83]. Additionally, impedimetric measurements are physiologically non-specific. Pathophysiological changes in the VCA can arise from a plurality of possible overlapping sources. This requires the use of the proper experimental referencing conditions and controls. Lastly, the choice of constant current (CC) or constant voltage (CV) method must be addressed. Constant current approaches such as the enhanced Howland pump [84] are commonly used to limit current for TUT safety and reliably address variable electrode-to-TUT (E2TUT) contact impedance [85]. However, constant voltage techniques have been found to offer higher bandwidths [86]. Despite these challenges, BI/BIS has been implemented in several clinical applications, including the detection and monitoring of edema.

## 3. Summary and Perspective

Vascularized composite allotransplantation is a complex and multifaceted surgical procedure that may significantly improve a patient’s quality of life. The tissue composition of VCA complicates this issue due to the presence of highly damaged-susceptible muscle and nerve tissue. As further research seeks to develop methods to minimize graft damage while sustaining viability for extended periods of time, advancements in status monitoring technology during preservation will determine their efficacy. Our perspective is that there is a compelling need for the development of an inflammation status score for VCA during preservation. There is also a compelling need for the development of continuous monitoring technologies, exampled by MMBIS. Both, in combination with AI, will allow robust stratification of VCAs and enable the proper management of transplant rejection via quality stratification of allografts. Pre-transplant stratification of allografts accommodates early recognition and possible prevention of acute rejection episodes. MMBIS may then be used as a clinical surrogate during transportation, allowing wider distribution of allografts. Bioimpedance spectroscopy is a promising development that enables quantitative, real-time, rapid, and continuous tissue monitoring with spatiotemporal resolution [87]. While MMBIS may be fused with bioanalytical data reflective of the pathophysiology of the VCA and expert data to stratify pre-transplant rat abdominal wall, it is cautioned that such findings may not be directly transferable to humans. Moreover, while we have focused exclusively on pre-transplant stratification of VCA, it should be noted that post-transplant monitoring may also be feasible.

Limitations to bioimpedance applications in VCA include the lack of standardization in protocols, as well as unknown baselines for parameters measured. These include the ideal electrode placement, frequencies for measuring VCA impedance, and magnitude and phase threshold of impedance values in varying tissue types. With VCA, the anisotropic properties of the skin and tissues must be considered since the flow of electric current inside the human body is altered by the components and orientation of cells. The tetrapolar configuration of electrodes is most commonly used in bioimpedance measurements but is affected by the anisotropic properties of nerve fibers, muscle, and blood vessels [88]; consequently, new electrode configurations that mitigate their effects should be investigated [89]. While there has not been a significant implementation of MMBIS technology in the field of VCA, and there remain challenges in method standardization and data analysis, the hardware capabilities of this method present an opportunity to tremendously advance a field whose technological growth has stagnated for the past few decades. 

## Figures and Tables

**Figure 2 bioengineering-10-00434-f002:**
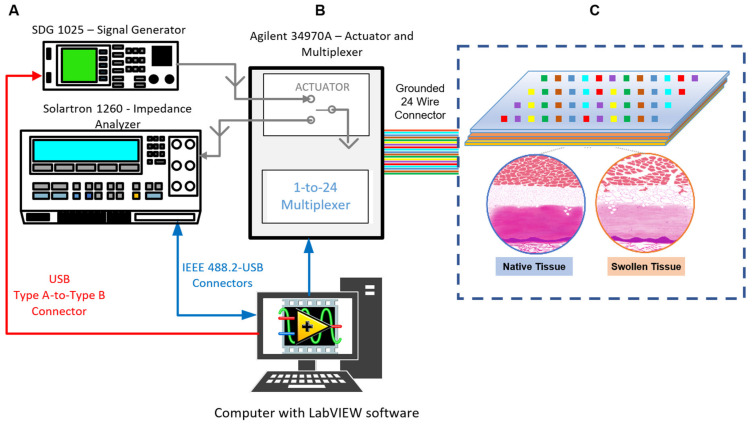
The use of multiplexed, multi-electrode, bioimpedance spectroscopy (MMBIS) to study the temporal changes in electrical properties of a vascularized composite allograft. (**A**) Instrumentation comprises a Frequency Response Analyzer (FRA) for multi-frequency impedance spectral monitoring, a Signal Generator for concomitant electrical stimulation of the allograph, and a multiplexer for independently addressing an array of deployed electrodes. (**B**) A schematic illustration of the application of MMBIS to an electrified vascularized composite allograft showing an electrode array deployed above (color-coded to the indexed array) and below the allograft (Adapted with permission from [56]. 2021, Elsevier). (**C**) A schematic illustration of the physiological basis for a change in bioimpedance based on swelling (edema) of the allograft. Native or un-swollen tissue is shown within the left circle, whereas swollen tissue with an accumulation of interstitial fluid is shown within the right circle.

**Figure 3 bioengineering-10-00434-f003:**
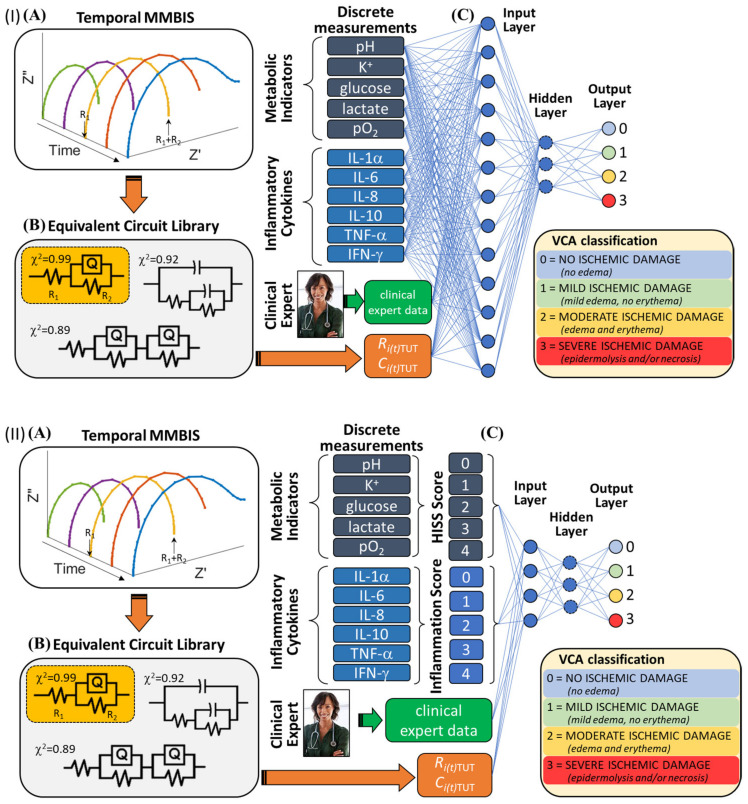
The use of temporal multiplexed, multielectrode, bioimpedance spectroscopy (MMBIS) data (**A**), rationalized through an equivalent circuit library (**B**) to yield discrete values of allograft resistance and capacitance (**C**) which, along with bioanalytical/bioassay data of key metabolites in Hemorrhage Intensive Severity and Survivability (HISS) scoring, key pro- and small anti-inflammatory molecules that enable an inflammation score, and clinical expert data may be fed into a suitable Artificial Neural Network (ANN) classifier to yield VCA classification corresponding to 0 = NO ISCHEMIC DAMAGE (no edema), 1 = MILD ISCHEMIC DAMAGE (mild edema, no erythema), 2 = MODERATE ISCHEMIC DAMAGE (edema and erythema), and 3 = SEVERE ISCHEMIC DAMAGE (epidermolysis and/or necrosis). (**I**) Discrete measurements passed to the ANN classifier, or (**II**) derivative scores passed to the ANN classifier.

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
