# Peer review of "Real-Time Monitoring Using Multiplexed Multi-Electrode Bioelectrical Impedance Spectroscopy for the Stratification of Vascularized Composite Allografts: A Perspective on Predictive Analytics"

_bioengineering, 2023, doi:10.3390/bioengineering10040434_

Round 1

Reviewer 1 Report

This article is a very academic and impressive report, showing us a great perspective.  The manuscript is of clinical significance, needs minor revisions 

: If allograft viability stratification prior to transplantation can be a quantitative fusion of the multi-modal bioanalytical/bioassay and biophysical data with clinical expert data via machine learning/artificial intelligence algorithms, it would be great!  Also, if VCA stratification works in machine perfusion stage, why is that just for assessing graft preservation efficacy and viability?  MMBIS and ML fusion..   Could it be also for VCA post-transplant rejection and tolerance evaluation and scoring?  Is there any specific hurdle for VCA?  Some difference with other solid organ?

Author Response

Reviewer 1:

This article is a very academic and impressive report, showing us a great perspective.  The manuscript is of clinical significance, needs minor revisions: If allograft viability stratification prior to transplantation can be a quantitative fusion of the multi-modal bioanalytical/bioassay and biophysical data with clinical expert data via machine learning/artificial intelligence algorithms, it would be great!  Also, if VCA stratification works in machine perfusion stage, why is that just for assessing graft preservation efficacy and viability? MMBIS and ML fusion. Could it be also for VCA post-transplant rejection and tolerance evaluation and scoring? Is there any specific hurdle for VCA? Some difference with other solid organ?

We thank the reviewer for these insightful questions. Indeed, we believe that MMBIS, once developed for allograft stratification during preservation, may be extended to post-transplantation monitoring of rejection. VCA, unlike solid organs, are particularly challenging because of the considerable tissue heterogeneity of the allograft. Moreover, when compared to solid organ transplantation, acute rejection rates are approximately six-times greater. This necessitates vastly different pre-transplantation protocols with aggressive immunosuppression, and quite often, loss of graft viability. Where mentioned in the text, we have modified the sentences to emphasize these two points of distinction between solid organs and allografts.

VCA, unlike solid organs, are particularly challenging because of the considerable tissue heterogeneity of the allograft. Moreover, when compared to solid organ transplantation, acute rejection rates are approximately six-times greater. This necessitates vastly different pre-transplantation protocols with aggressive immunosuppression, and quite often, loss of graft viability. The applicability of fused MMBIS biophysical data, bioanalytical physiological data and clinical expert data once successfully applied to preserved allografts may potentially be extended to post-transplant rejection and tolerance evaluation and scoring. 

Reviewer 2 Report

As compared to a recent review of the Authors (Bioelectrical Impedance Spectroscopy for Monitoring Mammalian Cells and Tissues under Different Frequency Domains: A Review by Sara Abasi et al. (ACS Meas. Sci. Au 2022, 2, 6, 495–516), the current manuscript presents a perspective view on the use of Multiplexed Multi-electrode Bioelectrical Impedance Spectroscopy (MMBIS) as a suitable technique for the assessment of „clinically relevant edema” formation in a very specific medical condition, i.e. vascularized composite allotransplantation (VCA). Besides, the Authors propose that „this method has the potential to provide the spatial and temporal resolution required for continual VCA monitoring during preservation”.

Bioelectrical impedance analysis was introduced several decades ago to measure the fluid status of the body in clinical conditions. Although not extensively researched, there are some reports related to the use of bioimpedance to detect ischemia-induced edema formation too, at least in solid organs in transplantation model experiments. The recognition of edema formation in VCA settings is a novel area, it would lead us directly to the development of specific diagnostics and/or therapeutics and can surely have profound and far-reaching impacts in human transplantation medicine.  In this respect this work is timely, and the field can benefit by receiving a new aspect from an expert perspective. Nevertheless, there are some concerns and missing information that limit the enthusiasm of the reviewer.

1. If we accept the presumption that fluid accumulation within stored and artificially perfused tissue beds is a key indicator of VCA viability then the use of proper control conditions is needed to obtain the referencing baselines.  In this respect, the complex anatomical structures of VCA which contain multiple tissue types including fat, muscle, connective tissue and so on can confound bioimpedance parameters - fat cells have proportionally lower intracellular water. If such data exist, i.e. detection of the dynamics of oedema formation by bioelectrical impedance spectroscopy in solid fat-containing organs prepared for clinical or experimental transplantation, these should be cited.

2.  VCA is a relatively new field of surgery that involves reconstruction of composite tissues (such as the hand or face) in one procedure. The success is not guaranteed, however, mostly due to the extremely antigenic properties of certain standard components of the graft (skin, nerves, etc) and the increased level of different immune responses compared to solid organ transplants. Are there any previous reports on edema formation in VCA and diagnosed by bioimpedance-based analysis? Of note, a minimum amount of scientific evidence is needed before it is proposed that „when combined with AI, MMBIS may enable prognostic stratification of VCA to increase successful outcomes following transplantation.”

3. Why do you want to use the ex vivo machine perfusion system?  This is a highly ambitious goal, because testing could be much easier in static cold storage conditions.  How to link edema-associated electrical behaviour with other pathophysiological changes in the fluid filled tissues - in other words, how to prove specificity? Please define a reference method in determining graft edema in machine perfused (fluid filled) VCA transplant population.

To sum up, the available evidence suggest that this method can indeed provide useful information in animal models, but this situation is not equal to static- or machine-perfused clinical conditions. The in vitro or in vivo results should always be evaluated cautiously, and usually cannot be translated directly to a human scenario. Certain logical links, including scientific evidence, which could support the use of bioimpedance spectroscopy for the diagnosis of intragraft edema formation and prognostic stratification of VCA in humans are missing, so explicit statements referring to clinical conditions should be replaced by more restricted ones in the abstract, introduction and discussion.

Minor

The method itself (Multiplexed Multi-electrode Bioelectrical Impedance Spectroscopy) should be included in the title.

Author Response

Reviewer 2:

As compared to a recent review of the Authors (Bioelectrical Impedance Spectroscopy for Monitoring Mammalian Cells and Tissues under Different Frequency Domains: A Review by Sara Abasi et al. (ACS Meas. Sci. Au 2022, 2, 6, 495–516), the current manuscript presents a perspective view on the use of Multiplexed Multi-electrode Bioelectrical Impedance Spectroscopy (MMBIS) as a suitable technique for the assessment of „clinically relevant edema” formation in a very specific medical condition, i.e. vascularized composite allotransplantation (VCA). Besides, the Authors propose that „this method has the potential to provide the spatial and temporal resolution required for continual VCA monitoring during preservation”. 

Bioelectrical impedance analysis was introduced several decades ago to measure the fluid status of the body in clinical conditions. Although not extensively researched, there are some reports related to the use of bioimpedance to detect ischemia-induced edema formation too, at least in solid organs in transplantation model experiments. The recognition of edema formation in VCA settings is a novel area, it would lead us directly to the development of specific diagnostics and/or therapeutics and can surely have profound and far-reaching impacts in human transplantation medicine.  In this respect this work is timely, and the field can benefit by receiving a new aspect from an expert perspective. Nevertheless, there are some concerns and missing information that limit the enthusiasm of the reviewer.

We thank the reviewer for this positive assessment of our perspective. We have addressed the concerns in our point-by-point responses below:

  1. If we accept the presumption that fluid accumulation within stored and artificially perfused tissue beds is a key indicator of VCA viability then the use of proper control conditions is needed to obtain the referencing baselines. In this respect, the complex anatomical structures of VCA which contain multiple tissue types including fat, muscle, connective tissue and so on can confound bioimpedance parameters - fat cells have proportionally lower intracellular water. If such data exist, i.e. detection of the dynamics of oedema formation by bioelectrical impedance spectroscopy in solid fat-containing organs prepared for clinical or experimental transplantation, these should be cited.

We thank the reviewer for this comment and for the recommendation to add additional citations that specifically address edema within specific tissue types, e.g., “solid fat-containing organs”. Interestingly, we have found no supporting literature on edema of specific tissue type(s). However, the bioimpedance and electrical conductivity of different tissue types is well studied and is represented in Figure 1D) Conductivity of different tissue over a frequency range of 10Hz-100GHz. The graph is reconstructed using data from Gabriel et al.38 Additionally, there is literature on the bioimpedance of the environment of solid organs, which was not intended to be covered by this perspective. Based on the suggestion of the reviewer, we have added the following additional references.

Zink et al. (Zink et al. 2020) compared body segmented BIS data with traditional clinical diagnostics in a study of fluid accumulation and status during cardiac recompensation. Because clinical signs of volume overload, such as breathlessness and peripheral edema, vary widely, the authors compared “whole-body” and “foot-to-foot” bioimpedance values, which were lowered for cases of peripheral edema, and “hand-to-hand” and “transthoracic” bioimpedance, which were lowered for cases of central edema.

Zink, M.D., König, F., Weyer, S. et al. Segmental Bioelectrical Impedance Spectroscopy to Monitor Fluid Status in Heart Failure. Sci Rep 10, 3577 (2020). https://doi.org/10.1038/s41598-020-60358-y

Edwick et al. (Edwick et al. 2020) has demonstrated the use of BIS as a valid, reliable, and sensitive technique in the measurement of extracellular fluid volumes, and hence edema, following acute hand burn injury. Like our purpose in VCA monitoring, they emphasized the importance of changes in edema to augment clinical decision making. The authors extracted the impedance of the intracellular component from a fitted Cole model to calculate the extracellular fluid (ECF) volume. These values were then compared with values obtained from the water displacement method, where the swollen hand was placed in a liquid bath and the volume displaced was measured, resulting in a Pearson’s correlation of r=0.793 (P<0.001).

Dale O Edwick, MPhty, Dana A Hince, PhD, Jeremy M Rawlins, MBBS, Fiona M Wood, MBBS, PhD, Dale W Edgar, PhD, Bioimpedance Spectroscopy Is a Valid and Reliable Measure of Edema Following Hand Burn Injury (Part 1—Method Validation), Journal of Burn Care & Research, Volume 41, Issue 4, July/August 2020, Pages 780–787, https://doi.org/10.1093/jbcr/iraa071

Crescenzi et al. (Crescenzi et al. 2019) used BIS to assess the temporal accumulation of tissue water and were able to distinguish lipedema from Dercum’s Disease with emphasis on identifying the early stages of lipedema. The authors found a reduced R0 of the Cole model to be associated with the higher volume of swollen legs of patients with more severe lipedema. This corresponded with the clinical manifestations of pitting upon palpation, and it was assumed that this was the result of a greater accumulation of fluid within the interstitial spaces of the tissues. The reduced impedance and increased interstitial fluid volume was suggested to occur because of an increase in sodium and associated counter anion content in the interstitial space (Crescenzi et al. 2018).

Rachelle Crescenzi, Paula M C Donahue, Sandra Weakley, Maria Garza, Manus J Donahue, Karen L Herbst, Lipedema and Dercum's Disease: A New Application of Bioimpedance Lymphat Res Biol (2019) 17(6):671-679. doi: 10.1089/lrb.2019.0011. Epub 2019 Aug 13. https://www.ncbi.nlm.nih.gov/pmc/articles/PMC6919257/pdf/lrb.2019.0011.pdf

Crescenzi R, Marton A, Donahue PMC, Mahany HB, Lants SK, Wang P, Beckman JA, Donahue MJ, Titze J. Tissue Sodium Content is Elevated in the Skin and Subcutaneous Adipose Tissue in Women with Lipedema. Obesity (Silver Spring). 2018 Feb;26(2):310-317. doi: 10.1002/oby.22090. Epub 2017 Dec 27. PMID: 29280322; PMCID: PMC5783748.

  1. VCA is a relatively new field of surgery that involves reconstruction of composite tissues (such as the hand or face) in one procedure. The success is not guaranteed, however, mostly due to the extremely antigenic properties of certain standard components of the graft (skin, nerves, etc) and the increased level of different immune responses compared to solid organ transplants. Are there any previous reports on edema formation in VCA and diagnosed by bioimpedance-based analysis? Of note, a minimum amount of scientific evidence is needed before it is proposed that „when combined with AI, MMBIS may enable prognostic stratification of VCA to increase successful outcomes following transplantation.”

We thank the reviewer of the accurate summary of the present status of VCA and for the question regarding previous studies on edema formation in VCA and its diagnosis by bioimpedance. As the reviewer has noted, the authors have done a comprehensive review of the literature [Bioelectrical Impedance Spectroscopy for Monitoring Mammalian Cells and Tissues under Different Frequency Domains: A Review by Sara Abasi et al. (ACS Meas. Sci. Au 2022, 2, 6, 495–516)] before advancing the present perspective view on the use of Multiplexed Multi-electrode Bioelectrical Impedance (MMBIS) in VCA monitoring. The authors found no instance of BIS being used to monitor edema formation in VCA during preservation. However, as noted above, there are several instances of bioimpedance being used to monitor solid organs such as rat kidneys (Genescà et al. 2005), and near homogeneous tissue organs such as the lungs and edema associated diseases such as Lipedema (discussed above). To illustrate the role of bioimpedance in organ and tissue monitoring, while not intending to be a repeat of our recent review, we have added the following.

In studying the preserved and transplanted solid organ rat kidney model, Genescà et al. (Genescà et al. 2005) used the Cole model parameters to clearly differentiate between two groups, distinguished by their BIS behavior during reperfusion. They identified the α parameter, the width of the bioimpedance dispersion, as a feature that is both objective and easy to allow real-time, continuous monitoring to detect and monitor ischemic damage during rat kidney preservation and “could be a promising method to assess graft viability in the future”. Additionally, τ, the characteristic time constant of the dispersion, was used to differentiate between the two groups during the short preservation times, ~ 24h. Both α and τ implicate the increased volume of ischemic cells for its impact on reduced interstitial volume and increased cell membrane surface area.

Meritxell Genescà, Antoni Ivorra, Anna Sola, Luis Palacios, Jean-Michel Goujon, Thierry Hauet, Rosa Villa, Jordi Aguiló, Georgina Hotter, Electrical bioimpedance measurement during hypothermic rat kidney preservation for assessing ischemic injury Biosensors and Bioelectronics, Volume 20, Issue 9, 2005, Pages 1866-1871, ISSN 0956-5663, https://doi.org/10.1016/j.bios.2004.06.038.

https://vimeo.com/566644747 (Predatory Journal Practices)

  1. Why do you want to use the ex vivo machine perfusion system? This is a highly ambitious goal, because testing could be much easier in static cold storage conditions.How to link edema-associated electrical behaviour with other pathophysiological changes in the fluid filled tissues - in other words, how to prove specificity? Please define a reference method in determining graft edema in machine perfused (fluid filled) VCA transplant population.

We thank the reviewer for the question regarding ex vivo machine perfusion of allografts prior to transplantation. This is not as ambitions as one may at first think. There are reports in the literature of allograft preservation bioreactors with perfusion. Our group has for several years been at the forefront in the design, development and application of multi-parametric bioreactors for the functional preservation of vascularized composite allografts (Salehi et al. 2018a) (Salehi et al. 2018b). The authors have also published on systems with ex vivo electrical stimulation to promote muscle regeneration and reduce muscle atrophy (Somers and Grayson 2021) (Somers et al. 2019) (Jun et al. 2022). To address this comment, we have included a brief synopsis of our prior work in this area.

Static cold storage has been shown to present major limitations for VCA (Burlage et al. 2018). In response to these limitations, our team has reported on the design, development and application of multi-parametric bioreactors for the functional preservation of vascularized composite allografts (Salehi et al. 2018a) (Salehi et al. 2018b). Such systems have been imbued with the capability to support ex vivo electrical stimulation to promote muscle regeneration and reduce muscle atrophy (Somers and Grayson 2021) (Somers et al. 2019) (Tran 2018)while monitoring physiological biomarkers (Gilbert-Honick and Grayson 2020) (Jun et al. 2022).

Salehi S, Tran K, Grayson WL. Focus: Medical technology: Advances in Perfusion Systems for Solid Organ Preservation. Yale J Biol Med. 2018 Sep 21;91(3):301-312. PMID: 30258317; PMCID: PMC6153619.

S Salehi, W Grayson, G Brandacher, G Furtmuller, J Lopez Establishing a rat abdominal wall perfusion model for VCA preservation Cryobiology (2018) 81, 232. https://doi.org/10.1016/j.cryobiol.2017.12.078

Sarah M. Somers and Warren L. Grayson “Protocol for the Use of a Novel Bioreactor System for Hydrated Mechanical Testing, Strained Sterile Culture, and Force of Contraction Measurement of Tissue Engineered Muscle Constructs”, Front. Cell Dev. Biol., 2021, 9: 815| https://doi.org/10.3389/fcell.2021.661036

Somers, Sarah M., Zhang, Nicholas Y., Morrissette-McAlmon, Justin B. F., Tran, Kenny, Mao, Hai-Quan, Grayson, Warren L. “Myoblast Maturity on Aligned Microfiber Bundles at the Onset of Strain Application Impacts Myogenic Outcomes”, Acta Biomaterialia, 2019, 94:232-242. https://doi.org/10.1016/j.actbio.2019.06.024

Jun I, Li N, Shin J, Park J, Kim YJ, Jeon H, Choi H, Cho JG, Chan Choi B, Han HS, Song JJ. Synergistic stimulation of surface topography and biphasic electric current promotes muscle regeneration. Bioact Mater. 2021 Oct 19;11:118-129. https://doi.org/10.1016/j.bioactmat.2021.10.015. PMID: 34938917; PMCID: PMC8665271.

How to link edema-associated electrical behaviour with other pathophysiological changes in the fluid filled tissues - in other words, how to prove specificity?

Our goal is not to define causal specificity, although, that may be an eventual reductionist objective. Our goal in our perspective to define a general approach towards VCA stratification. In this context, specificity is not relevant. However, our group has used several computational techniques, including Hierarchical Clustering Analysis (HCA), Principal Component Analysis (PCA), Random Forest Classification (RFC) and Multinomial Logistic Regression (MLR) models to identify main effect contributors to the inflammatory response of VCA during preservation. Our present approach that employs AI algorithms such as deep learning artificial neural networks, support vector machines, decision tree analysis, possibility analysis, etc. fuses pathophysiology analytical data, clinical expert data, and MMBIS data to yield stratified performance. The authors have employed similar approaches in the development of a Hemorrhagic Trauma Severity scoring system (Bhat et al. 2020) for the hemorrhaging trauma patient (Shickel et al. 2022). To address this comment, we have included the following statement to in our discussion.

While there may be interest in the underlying causes of edema and the molecular etiology of changes in the bioimpedance associated with the development of edema, the present perspective seeks only to wholistically stratify the VCA based on the fusion of pathophysiology analytical data, clinical expert data, and MMBIS data to yield stratified performance. The authors have employed similar approaches in the development of a Hemorrhagic Trauma Severity scoring system (Bhat et al. 2020) for the hemorrhaging trauma patient (Shickel et al. 2022).

  1. Ankita Bhat, Daria Podstawczyk, Brandon K. Walther, John R. Aggas, David Machado-Aranda, Kevin R. Ward, and Anthony Guiseppi-Elie “Toward a Hemorrhagic Trauma Severity Score: Fusing Five Physiological Biomarkers” Journal of Translational Medicine (2020) 18, 348 https://doi.org/10.1186/s12967-020-02516-4 (IF= 4.200)
  2. Benjamin Shickel, Jeremy Balch, John R. Aggas, Tyler J. Loftus, Christian N. Kotanen, Parisa Rashidi and Anthony Guiseppi-Elie* “Scoring for Hemorrhage Severity in Traumatic Injury” In: Rajendram, Rajkumar; Preedy, Victor R.; Patel, Vinood B. (eds) Biomarkers in Trauma, Injury and Critical Care. Biomarkers in Disease: Methods, Discoveries and Applications. Book Series Editors: Vinood B. Patel and Victor R. Preedy. Springer Nature Book Series, Springer Nature, New York. Electronic ISSN: 2542-3665; Print ISSN: 2542-3657. ISBN: 978-3-031-07394-6 https://doi.org/10.1007/978-3-030-87302-8_58-1

Please define a reference method in determining graft edema in machine perfused (fluid filled) VCA transplant population.

It is feasible to use other methods to measure edema. Among them, the resect and weigh to dry weight method or the fluid volume displacement method. The former is a highly invasive method that destroys both the sample and further traumatizes the VCA. The latter requires the allograft be “dunked” into a fluid bath and the volume displacement of fluid measured. Both methods are highly disruptive to a perfusion system. To address this request for a reference method, we have included the following statement to in our discussion.

There are other methods to measure edema. Among them, the resect and weigh to dry weight method or the fluid volume displacement method. The former is a highly invasive method that destroys both the sample and further traumatizes the VCA. The latter requires the allograft be “dunked” into a fluid bath and the volume displacement of fluid measured. Both methods are highly disruptive to a perfusion system.

In the work of Strand-Amundsen et al. (Strand-Amundsen et al. 2017) they used bioimpedance to study the temporal changes in the electrical properties of the multilayered small intestine, both in vivo nd ex vivo, during ischemia development. The authors found significant differences (p < 0.0001) between the evolving electrical parameters as a function of ischemic time development when comparing measurements performed in vivo vs. ex vivo. However, there were notable similarities in the trends, both trending downward, likely due to edema, with the ex vivo data showing a low frequency rise for about 3 h prior to beginning its downward trend. The temporal evolution of the phase angle, φ, was similarly significantly different (p  <  0.0001) when comparing the baseline condition and the development of ischemia in both the in vivo nd ex vivo, models. It is noteworthy that these differences are more strongly frequency dependent and typically occur in the 1 -7h time frame.

Peterson et al. (Peterson et al. 2022) used bioimpedance to study total organ edema of porcine lungs undergoing normothermic extravascular lung perfusion (EVLP). They used Steen, an asanguinous and hyperosmolar solution, normal saline (0.9%), and cell culture media to determine if bioimpedance was associated with vascular lung water and lung injury severity. Overall increases in pulmonary edema and extravascular lung water have a direct negative correlative effect on lung organ quality. Steen is known for its protective effect on the pulmonary microvascular endothelium (Ta et al. 2023). Saline and Cell culture media both induced pulmonary edema and lung injury. Steen, on the other hand reduced lung injury and extravascular free water over the duration of normothermic EVLP. They found that bioimpedance could provide a quantitative organ assessment that would allow for more accurate pre-transplant clinical decisions.

In the recent work of Hou et al (Hou et al. 2021) they studied the temporal bioimpedance of resected segments of the highly layered human small intestine. The found that the normalized resistance ((R0-Ri)/R0)=Py) initially decreased and subsequently increased before decreasing over a 10h period of monitoring. The time interval where Py value returned to its maximum value was found to be consistent with reported viable/non-viable limits based on histological analysis. In this case ischemia was associated with the internal swelling of cells and changes in the ratio between intracelluar and extracelluar fluids, edema. In related work of Hou et al. (Hou et al. 2022), dielectric relaxation spectroscopy was used to assess intestinal viability of healthy, ischemic and re-perfused segments of intestine in a porcine mesenteric ischemia model. The bioelectric parameters of dielectric constant and conductivity showed clear differences between healthy, ischemic and re-perfused intestinal segments. This confirmed the value of dielectric parameters in characterizing different intestinal conditions likely based on edema. They then employed machine learning models to classify viable and non-viable segments based on frequency dependent dielectric properties of the intestinal segment. The combined approach of dielectric relaxation and trained machine learning is speculated to allow the surgeon to make accurate evaluation by performing a single measurement on an intestinal segment where the viability state is questionable. This then augments the surgeon’s difficult decision regarding resection segments and resection margins. Moreover, the early stages of the temporal changes in bioimpedance measured during ischemia produces a pattern of edema that is different in porcine and human small intestine.

Runar J Strand-Amundsen, Henrik M Reims, Christian Tronstad, Håvard Kalvøy, Ørjan G Martinsen, Jan O Høgetveit, Tom E Ruud and Tor I Tønnessen Ischemic small intestine—in vivo versus ex vivo bioimpedance measurements Physiological Measurement (2017) 38(5) 715 DOI 10.1088/1361-6579/aa67b7

Danielle M. Peterson, Eliza W. Beal, Brenda F. Reader, Curtis Dumond, Sylvester M. Black, and Bryan. A. Whitson Electrical Impedance as a Noninvasive Metric of Quality in Allografts Undergoing Normothermic Ex Vivo Lung Perfusion ASAIO J. 2022; 68(7): 964–971. doi: 10.1097/MAT.0000000000001591

Ta HQ, Teman NR, Kron IL, Roeser ME, Laubach VE. Steen solution protects pulmonary microvascular endothelial cells and preserves endothelial barrier after lipopolysaccharide-induced injury. J Thorac Cardiovasc Surg. 2023 Jan;165(1):e5-e20. doi: 10.1016/j.jtcvs.2022.04.005. Epub 2022 Apr 18. PMID: 35577593; PMCID: PMC9576825.

Hou J, Strand-Amundsen R, Hødnebø S, Tønnessen TI, Høgetveit JO. Assessing Ischemic Injury in Human Intestine Ex Vivo with Electrical Impedance Spectroscopy. J Electr Bioimpedance. 2021 Nov 29;12(1):82-88. doi: 10.2478/joeb-2021-0011

Hou, J., Strand-Amundsen, R., Tronstad, C. et al. Small intestinal viability assessment using dielectric relaxation spectroscopy and deep learning. Sci Rep 12, 3279 (2022). https://doi.org/10.1038/s41598-022-07140-4

To sum up, the available evidence suggest that this method can indeed provide useful information in animal models, but this situation is not equal to static- or machine-perfused clinical conditions. The in vitro or in vivo results should always be evaluated cautiously, and usually cannot be translated directly to a human scenario. Certain logical links, including scientific evidence, which could support the use of bioimpedance spectroscopy for the diagnosis of intragraft edema formation and prognostic stratification of VCA in humans are missing, so explicit statements referring to clinical conditions should be replaced by more restricted ones in the abstract, introduction and discussion.

We thank the reviewer for this comment. We have added a cautionary note regarding performance in animal models, such as rat abdominal wall, and the transfer of such findings to humans.

While MMBIS may be fused with bioanalytical data reflective of the pathophysiology of the VCA and expert data to stratify pre-transplant rat abdominal wall, it is cautioned that such finding may not be directly transferable to humans. Moreover, while we have focused exclusively on pre-transplant stratification of VCA, it should be noted that post-transplant monitoring may also be feasible.

Minor

The method itself (Multiplexed Multi-electrode Bioelectrical Impedance Spectroscopy) should be included in the title.

We have updated the title accordingly.“Real-Time Monitoring using Multiplexed Multi-electrode Bioelectrical Impedance Spectroscopy for the Stratification of Vascularized Composite Allografts: A Perspective on Predictive Analytics”

Reviewer 3 Report

Overall, the manuscript is reasonably well put together. There are two minor issues-

1) Parts of the abstract are difficult to understand and do not quite encapsulate the summary of where this particular technology is currently at with respect to being used in VCA transplantation (either for ongoing research and/or clinically). Specifically - what do you mean in the 3rd sentence of the abstract -are you trying to say that prolonged cold ischemia is negatively correlated with VCA transplantation outcomes, or do you mean something else entirely? It appears that BIS as yet is not established as a validated tool for monitoring of VCA either prior to or post transplantation-at this stage it is only a possible option (if a number of the limitations with the current available technology are addressed and overcome). This needs to be made a lot clearer in the abstract.

2) With respect to the manuscript. Can you make it clearer if there is any potential for the currently available BIS devices to be trialed as part of ongoing VCA research (including in animal models) and/or on human VCA which are deemed not suitable for transplantation or does in fact a completely new type of MMBIS device now need to be developed? You need to expand on what research also now needs to be undertaken in VCA animal models etc. pertaining to also developing an inflammation status score as well.

Author Response

Reviewer 3:

Overall, the manuscript is reasonably well put together. There are two minor issues-

  • Parts of the abstract are difficult to understand and do not quite encapsulate the summary of where this particular technology is currently at with respect to being used in VCA transplantation (either for ongoing research and/or clinically). Specifically - what do you mean in the 3rd sentence of the abstract -are you trying to say that prolonged cold ischemia is negatively correlated with VCA transplantation outcomes, or do you mean something else entirely? It appears that BIS as yet is not established as a validated tool for monitoring of VCA either prior to or post transplantation-at this stage it is only a possible option (if a number of the limitations with the current available technology are addressed and overcome). This needs to be made a lot clearer in the abstract.

We thank the reviewer for these comments. Indeed, the reviewer’s observations are correct in both instances. i) prolonged cold storage ischemia is negatively correlated with VCA transplantation outcomes, and ii) BIS is not yet established as a validated tool for monitoring VCA, either prior to or post-transplantation, because of the limitations and opportunities discussed on this perspective.

We have modified the abstract to clearly state these points.

  • With respect to the manuscript. Can you make it clearer if there is any potential for the currently available BIS devices to be trialed as part of ongoing VCA research (including in animal models) and/or on human VCA which are deemed not suitable for transplantation or does in fact a completely new type of MMBIS device now need to be developed? You need to expand on what research also now needs to be undertaken in VCA animal models etc. pertaining to also developing an inflammation status score as well.

We thank the reviewer for these suggestions that would help to clarify and strengthen our perspective. In our recent review of the bioimpedance technique and instrumentation, we provided a comprehensive table detailing the presently available bioimpedance instrumentation of clinical and research significance (Bioelectrical Impedance Spectroscopy for Monitoring Mammalian Cells and Tissues under Different Frequency Domains: A Review by Sara Abasi et al. (ACS Meas. Sci. Au 2022, 2, 6, 495–516). To enhance this aper, we have identified three commercially available BIS systems that possess some aspects of features that will allow them to be used in preliminary research of this perspective. However, uniquely developed MMBIS must be used to address the specific challenges of allograft stratification.

Commercially available, USFDA cleared bioimpedance devices and systems have been developed and deployed for clinical use (Abasi et al. 2022). Amongst them are the Baxter-toSense’s CoVa Monitoring System 2, a remote patient monitoring necklace that tracks vitals and cardiac fluids relevant in congestive heart failure (CHF), chronic obstructive pulmonary disease (COPD), hypertension, and renal failure. Cardiac fluids are tracked by bioimpedance plethysmography (10 kHz to 500 kHz) and serves as an early indicator of heart failure. The Philips Biosensor BX100 is a wearable system deployed as a single-use, 5-day wearable patch that uses thoracic bioimpedance (20kHz to 100kHz) to monitor the respiration of patients. The uCor3 by ZOLL Medical Corp. is similarly deployed as a patch for thoracic RF bioimpedance (0.5-2.5 GHz) for heart failure monitoring.

https://www.usa.philips.com/a-w/about/news/archive/standard/news/press/2020/20200526-philips-launches-next-generation-wearable-biosensor-for-early-patient-deterioration-detection-including-clinical-surveillance-for-covid-19.html

https://www.mobihealthnews.com/content/tosenses-remote-patient-monitoring-necklace-gets-fda-clearance-measure-stroke-volume-and

https://fccid.io/2ABHFUCOR30/User-Manual/Manual-3327234

Round 2

Reviewer 2 Report

Thank you for the bona fide reply. The ms imroves substantially, no further comments.

Author Response

Thank you!

Reviewer 3 Report

The manuscript reads a lot better in light of the revisions which have been undertaken by the authors. Of note there appear to be some formatting issues which may or may not be resolved by the authors accepting all of the tracked changes for this manuscript. Can the authors please check the following-

1) That Figure 1 is depicted accurately

2) That the (Insert the two equations) which appears beneath the second paragraph on page 6 is addressed

3) That Figure 3 is in the appropriate format

4) That the equation which appears in large black font on page 14 is now in the appropriate font size and appears in the text at the appropriate point in the manuscript

Author Response

The manuscript reads a lot better in light of the revisions which have been undertaken by the authors. Of note there appear to be some formatting issues which may or may not be resolved by the authors accepting all of the tracked changes for this manuscript. Can the authors please check the following-

  • That Figure 1 is depicted accurately

Figure 1 has been correctly depicted.

  • That the (Insert the two equations) which appears beneath the second paragraph on page 6 is addressed.

The equations beneath the second paragraph on page 6 have been inserted and all equations renumbered.

  • That Figure 3 is in the appropriate format.

Figure 3 is in the appropriate format.

  • That the equation which appears in large black font on page 14 is now in the appropriate font size and appears in the text at the appropriate point in the manuscript.

The font of the equation on page 14 has been corrected.